# Genetic Determinants of Macrolide and Fluoroquinolone Resistance in *Mycoplasma genitalium* and Their Prevalence in Moscow, Russia

**DOI:** 10.3390/pathogens12030496

**Published:** 2023-03-22

**Authors:** Inna Alexandrovna Edelstein, Alexandr Evgenjevich Guschin, Andrew Vyacheslavovich Romanov, Ekaterina Sergeevna Negasheva, Roman Sergeevich Kozlov

**Affiliations:** 1Institute of Antimicrobial Chemotherapy, Smolensk State Medical University of the Ministry of Health of the Russian Federation, 214019 Smolensk, Russia; andrew.romanov@antibiotic.ru (A.V.R.); roman.kozlov@antibiotic.ru (R.S.K.); 2Moscow Scientific and Practical Center of Dermatovenerology and Cosmetology, 125008 Moscow, Russia; aguschin1965@mail.ru (A.E.G.); alfo4ka@inbox.ru (E.S.N.)

**Keywords:** *Mycoplasma genitalium*, polymerase chain reaction (PCR), macrolides, fluoroquinolones, antibiotic resistance

## Abstract

Macrolide (MLR) and fluoroquinolone (FQR) resistance in *Mycoplasma genitalium* (MG) has recently become a major problem worldwide. The available data on the prevalence of MLR and FQR in MG in Russia are limited. In this study, we aimed to evaluate the prevalence and pattern of mutations in 213 MG-positive urogenital swabs from patients in Moscow between March 2021 and March 2022. MLR- and FQR-associated mutations were searched in 23S rRNA as well as in the parC and gyrA genes using Sanger sequencing. The prevalence of MLR was 55/213 (26%), with A2059G and A2058G substitutions being the two most common variants (36/55, 65%, and 19/55, 35%, respectively). FQR detection showed 17% (37/213); two major variants were D84N (20/37, 54%) and S80I (12/37, 32.4%) and three minor variants were S80N (3/37, 8.1%), D84G (1/37, 2.7%), and D84Y (1/37, 2.7%). Fifteen of the fifty-five MLR cases (27%) simultaneously harbored FQR. This study revealed the high frequency of MLR and FQR. We conclude that the improvement of patient examination algorithms and therapeutic approaches should be combined with the routine monitoring of antibiotic resistance based on the sensitivity profiles presented. Such a complex approach will be essential for restraining the development of treatment resistance in MG.

## 1. Introduction

The World Health Organization (WHO) project entitled “Global health sector strategies on, respectively, HIV, viral hepatitis and sexually transmitted infections for the period 2022–2030” defines the problem of increasing and spreading resistance of infectious organisms to antimicrobial agents as a high-priority one problem, and much attention is devoted to the development of appropriate measures aimed at controlling and restraining this resistance [1]. This area of research is focused on studying obligate pathogens capable of quick resistance development and its expansion through the population. *Mycoplasma genitalium* is one of such microorganisms. Its certain biological features (slow growth rate in cultivation media and the absence of a cell wall) dictate the usage of molecular biology techniques that detect specific fragments of DNA and/or RNA to perform quick diagnosis verification and investigation of molecular mechanisms of resistance. The causal treatment of mycoplasmal infection of the urogenital system employs antibacterial agents that inhibit protein biosynthesis (tetracyclines and macrolides) or DNA replication (fluoroquinolones) [2,3,4]. The national guidelines on the treatment of infections such as non-gonococcal male urethritis, cervicitis, vaginitis, endometritis, and pelvic inflammatory disorders, which are caused by *M. genitalium*, instruct to administer eradication therapy with these antibiotic compounds for at least 10 days. Such regimens are supposed to reduce the chance of resistant strain emergence by sustaining a steady antibiotic concentration in the patient’s tissues [5,6,7,8]. Nevertheless, *M. genitalium* resistance to macrolide and fluoroquinolone agents is currently a growing problem, as it can complicate choosing the optimal patient treatment strategy and therefore paves the way for new clinical therapeutic guidelines based on new discoveries in the field [9,10,11,12,13].

One of the mechanisms of antimicrobial resistance in *M. genitalium* is the occurrence of DNA mutations in the target genes. Genetic changes resulting in macrolide resistance are primarily represented by a single-base mutations in either the A2058 or the A2059 position, or very rarely in A2062 (*E. coli* numbering) in the V domain region of the 23S rRNA gene [3]. Fluoroquinolone resistance arises from amino acid substitutions in quinolone resistance determining regions (QRDRs) in DNA gyrase GyrA or topoisomerase IV subunit ParC [14]. Mutations in the *gyr*A locus for *M. genitalium* are statistically less frequent in the published sample sets, and their impact upon fluoroquinolone sensitivity reduction in bacteria is less prominent in comparison to *parC* mutations [15]. *M. genitalium* isolates that are resistant to fluoroquinolones mostly display ParC amino acid substitutions in the S80 and D84 positions (*E. coli* numbering) [16,17]. A lack of therapeutic effect and bacterial eradication after replacement of macrolide treatment with fluoroquinolone and vice versa is reported in patients with strains demonstrating confirmed resistance to both antibiotics [18,19].

Therapy failure is usually linked to primary infection with strains that have already developed a resistance-associated mutation profile due to previous antibiotic exposure (poor personal compliance of the patient or a lack of a full recovery confirmation) or to the treatment-induced development of resistance, which is further transmitted through sexual contact [11]. The screening for *M. genitalium* infection is not performed routinely among asymptomatic patients, and the undefined microbial status may contribute to the spreading of mutant strains [6,12]. In the case of co-infection with *Chlamydia trachomatis*, empirical therapy with antibiotics can further boost resistance development and spread, because the antibiotic agents are prescribed with the prospect of concurrent elimination of two pathogens at once [8].

The implementation of efficient regimens for *M. genitalium* infection treatment and the development of a strategy to restrain the dissemination of resistance demand for more data to be collected across the world and in the Russian Federation in particular. A set of important measures to achieve these goals includes large-scale usage of commercially available assays for antibiotic resistance detection, the accumulation of confirmed data on resistance prevalence in different geographic regions and patient populations, and clinical monitoring of treatment efficacy. We aimed to evaluate the prevalence and nature of mutations providing the resistance to macrolide and fluoroquinolone antibiotics in *M. genitalium* samples isolated from patients examined at the Moscow Scientific and Practical Center of Dermatovenerology and Cosmetology (MSPCDC) between March 2021 and March 2022.

## 2. Materials and Methods

### 2.1. Experimental Design

The experimental collection included 213 DNA samples isolated from *M. genitalium*-positive cervical and urethral swab specimens obtained from patients examined at the Moscow Scientific and Practical Center of Dermatovenerology and Cosmetology (MSPCDC) and screened for sexually transmitted diseases (STD). DNA was isolated at a local lab using a “MagnoPrime-FAST” kit (NextBio LTD, Moscow, Russia); *M. genitalium* DNA presence was detected using a “AmpliPrime^®^ NCMT” PCR reagent multiplex kit according to the manufacturer’s instructions (NextBio LTD, Moscow, Russia). The commercial PCR assay kit allows for the simultaneous detection of *N. gonorrhoeae* DNA, *C. trachomatis* DNA, *M. genitalium* DNA, and *T. vaginalis* DNA. DNA samples were frozen and stored at −70 °C until they were sent to the Laboratory of Molecular Diagnostics at the Institute of Antimicrobial Chemotherapy, Smolensk State Medical University, Russia. The mutations associated with macrolide and fluoroquinolone resistance were identified using Sanger sequencing under the scope of the “DeMaRes” (Detection of Macrolide Resistance in *Mycoplasma genitalium*) investigation project (https://amrcloud.net/en/project/demares/, accessed on 3 February 2023). The whole sample set and the associated information were processed according to established protocols, and the experimental steps are depicted in Figure 1.

### 2.2. Validation Criteria

The present study included attendees of the Moscow Scientific and Practical Center of Dermatovenerology and Cosmetology (MSPCDC). Urethral swabs from males and cervical swabs from females were taken according to the established protocols. None of the patients received antibacterial treatment prior to testing. Three types of specimens were excluded from the analysis: (1) samples obtained from different anatomical regions of the same patient; (2) test-of-cure samples obtained after treatment (earlier than three weeks after treatment); and (3) recurrent positive samples taken after therapy changes.

### 2.3. Detection of M. genitalium and Macrolide and Quinolone Resistance

Detection of specific mutations linked to macrolide and fluoroquinolone resistance was performed via Sanger sequencing. The inner fragment of *M. genitalium* 23S rRNA and the QRDRs of the *par*C and *gyr*A genes (747, 470, and 279 base pairs, respectively) were amplified using specific primers. The corresponding PCR products were purified by exonuclease I, shrimp alkaline phosphatase treatment, and sequenced on both strands using the same primers and a BigDye^®^ Terminator v3.1 Cycle Sequencing Kit (Thermo Fisher Scientific, Waltham, MA, USA) on an Applied Biosystems 3500 Genetic Analyzer (Life Technologies, Carlsbad, CA, USA). DNA sequences were compared to reference sequence of the *M. genitalium* 23S rRNA gene (GenBank accession number NR_077054.1) or *parC* and *gyrA* QRDR sequences taken from the full *M. genitalium* genome (GenBank accession number NC_000908.2). The genes coordinated for the reported mutations are defined and presented in accordance with the currently accepted *E. coli* numeration system. The oligonucleotide primers used for fragment amplification are presented in Table 1.

### 2.4. Statistical Analysis

Data analysis was performed using the AMRcloud (https://amrcloud.net/en/) web platform and utilized standard methods of descriptive statistics: absolute and relative frequency calculation, median evaluation, estimation of Wilson 95% confidence intervals (CI), and multiple comparisons using Fisher’s exact test with Holm–Bonferroni correction [20].

## 3. Results

### 3.1. Specimens Involved in the Investigation

The majority of the specimens in the experimental set were obtained from male patients (170/213, 80%), while the cervical swab specimens contributed to 20% of the samples (43/213). The median age was 32.3 years for male patients (from 16 to 69 years) and 31.3 years for female patients (from 16 to 56 years).

### 3.2. Main Results of the Investigation

Our results demonstrate that 158 out of the 213 *M. genitalium* samples (74%; CI: 67.9–79.6%) have a wild-type genotype (WT) and were classified as “non-mutated sensitivity to macrolides”, while 55/213 samples (26%) tested positive for mutations associated with macrolide resistance (https://amrcloud.net/en/project/demares/, accessed on 3 February 2023). Further investigation of mutation types identified two variants of single-nucleotide substitutions in the *M. genitalium* 23S rRNA gene: the A2059G mutation was the prevalent mutation in the examined patient cohort (36/55 samples, 65%), while the second most common substitution was A2058G (19/55 samples, 35%). Fifteen of the fifty-five samples positive for macrolide resistance biomarkers (27%) displayed simultaneous amino acid substitutions in ParC, with them representing the cases with combinational resistance to two antibiotic families. In general, biomarkers of fluoroquinolone resistance were identified in 37/213 cases (17.4%), and the remaining 176 cases (82.6%; CI: 77–87.1%) displayed a WT genotype. Mutations associated with fluroquinolone resistance are represented by five variants of amino acid substitutions in ParC, with D84N and S80I being the most common ones observed in 20/37 (54%) and 12/37 (32.4%) cases, respectively. Other variants included very rare individual cases of D84G (1/37 case, 2.7%), D84Y (1/37 case, 2.7%), and S80N (3/37 cases, 8.1%). Locus 2611 of the *M. genitalium* 23S rRNA gene and QRDR of the *gyrA* gene did not have any mutations. The detailed results of the molecular biology analysis and genotype variant frequencies are presented in Table 2.

### 3.3. Additional Results of the Investigation

Considering the fact of frequent simultaneous detection of two obligate pathogens (*M. genitalium* and *Chlamydia trachomatis*) in the urogenital swabs, we additionally tested all available specimens for chlamydial DNA presence and found 13 out of the 213 samples (6%) to be positive. Five out of thirteen cases (38%) were also positive for the *M. genitalium* 23S rRNA gene mutations associated with macrolide resistance (four cases with A2058G and one case with A2059G substitutions). Mutations resulting in amino acid substitutions in ParC were detected in 6/13 samples (46%), with the D84N variant found in five samples and the S80I variant identified in one case. It is noteworthy that a combination of both resistance mechanisms was present in 3/13 samples (23%), which all displayed an A2058G+D84N substitution pattern.

## 4. Discussions

The predominant 23S rRNA mutation A2059G detected in 36/55 samples (65%) is likely to arise due to the administration of josamycin, a 16-membered macrolide that is used as a first line treatment agent for *M. genitalium* infections in the examined patient population. “Federal clinical guidelines on management of patients with urogenital disorders caused by *Mycoplasma genitalium*” recommend josamycin treatment as a therapy of choice for both simple and complicated forms of urogenital infections, including infections in pregnant women and children with a body weight less than 45 kg [21]. Studies describing the molecular mechanisms of josamycin’s interaction with the binding site in 23S rRNA demonstrate a strong correlation between josamycin’s molecular structure and A2059G mutation emergence which consequently results in increased minimum inhibitory concentrations of this agent for the tested strains in vitro [22,23]. Nineteen of the fifty-five samples (35%) also harbored A2058G nucleotide substitution, which also arises as a result of spontaneous mutations under macrolide therapy conditions. The total percentage of samples with A2059G and A2058G genotypes constituted 26%, and this value is higher than the one observed in a similar study performed at the same medical facility with a similar patient population in 2019, where 16% of the samples tested positive for mutations in 23S rRNA [24].

Large-scale epidemiological reviews and meta-analyses describe the spread of antibiotic resistance in *M. genitalium* in Europe [13,25]. Comprehensive reviews assess the interrelationships and differences in the management and treatment of sexually transmitted infections and the regional characteristics that account for the prevalence of antimicrobial resistance. Resistance to macrolides and fluoroquinolones is increasing in Russia compared with data published in 2015–2017. Research confirms the need for representative and well-defined surveillance of antimicrobial resistance in *M. genitalium*, both locally and at a European level.

Fluoroquinolone resistance in *M. genitalium* is usually caused by single amino acid substitutions in the GyrA and ParC proteins. The most common variants include D84N and S80I substitutions in the ParC protein (*E. coli* numbering) that alter the structure of the site targeted by the fluoroquinolone molecule and reduce its binding efficiency [26]. The fact that our data demonstrate that these two variants are the most common is in good agreement with observations reported in international studies [14]. S80N substitution, which is frequently detected in resistant *M. genitalium* strains, was experimentally proven to contribute to resistance development and therapy failure [27]. Other mutations are represented by single cases from our sample set and are rather uncommon in general; however, their connection to therapy inefficacy has been confirmed in a number of studies [28,29]. In comparison to data obtained in 2019, the frequency of fluoroquinolone resistance in the population increased from 14.5% to 17% [24].

Poorly controlled or inadequate use of fluoroquinolone agents with unconfirmed efficiency against *M. genitalium* may contribute to a significant increase in resistant strains’ prevalence. Two studies conducted in Japan in the early 2000s reported poor *M. genitalium* eradication efficacy and high recurrence chances after ofloxacin and levofloxacin treatment of non-gonococcal urethritis cases (31% and 50%, respectively) [30,31]. Treatment with 200 mg of ofloxacin twice a day for 10 days performed at the Unit for Sexual Transmitted Diseases in Oslo (Norway) did not result in *M. genitalium* elimination in 56% of cases [32]. The methodical guidelines published by Moscow City Healthcare Department in 2021 advise clinicians to avoid the administration of underperforming fluoroquinolones such as levofloxacin or ofloxacin because of the increasing prevalence of fluoroquinolone-resistant *M. genitalium* strains [33]. This increase may be attributed to a certain level of specificity in mutation selection by fluoroquinolones: second and third-generation agents directly target ParC in *M. genitalium*, therefore quickly promoting ParC-dependent resistance development. Moxifloxacin, a fourth-generation agent, has the same affinity toward both ParC and GyrA and thus retains high efficacy against *M. genitalium* strains resistant to ofloxacin and levofloxacin. A high antibacterial effect is facilitated through binding to the secondary target even if the primary one has mutated into a resistant variant [34,35].

The detection of 15 *M. genitalium*-positive samples (7%), which harbored the mutations providing simultaneous resistance to two families of therapeutic agents, is an important result of our study. The first report on such a phenomenon in the Russian Federation was published in 2017. A study devoted to the screening of *M. genitalium* resistance in four cities of the Russian Federation identified seven cases with combined resistance out of 659 cases in total (1.1%); however, more recent results reported in 2021 indicate that the prevalence of such isolates has increased up to 6.3% (17/268 cases) [36,37]. One of the plausible causes for the increase in the number of samples with such a genotype is selective pressure from macrolide and fluoroquinolone therapy due to the extensive use of these agents in both in-patient facilities and ambulatory treatment in recent years, especially in conjunction with anti-COVID-19 therapy [38]. The empirical prescription of these agents without an opportunity for timely STD diagnosis and a lack of therapy efficacy checks via confirmatory tests at the same time could promote the clonal expansion of such strains. A reasonable switch to fluoroquinolone therapy in the case of the poor efficacy of macrolides provided ground for the development of the secondary resistance to the new family of antibiotics and contributed to the stabilization and circulation of such isolates in the population [37]. Considering the statistically significant increase in the fraction of strains with such genotypes, the use of alternative therapeutic regimens or the introduction of new antimicrobial agents is necessary to overcome this spreading resistance. The definitive proof of a causal bond between different types of mutations identified in clinical *M. genitalium* samples and therapy failure may be obtained through the development of in vitro approaches to the evaluation of clinical isolates’ sensitivity to antimicrobial agents using classic culture techniques.

All proposed therapy variants should be based on efficient synchronous *C. trachomatis* and *M. genitalium* eradication and exclude monotherapy with doxycycline or one-time administration of azithromycin [18,22]. Unlike *C. trachomatis*, *M. genitalium* quickly develops resistance to the antibiotics of both first- and second-line treatment due to only one ribosomal operon being present in the genome [39]. Our data indicate that *M. genitalium* can already manifest fluoroquinolone resistance prior to initial macrolide treatment and vice versa; therefore, the ideal diagnostic approach should include testing for biomarkers associated with resistance to both agent groups.

Since *C. trachomatis* and *M. genitalium* were detected in different anatomical areas of the same patient (with extra-genital localization of the pathogen in the rectum and/or oropharynx), it is important to develop approaches for correction of diagnosis algorithms for patients from the population at risk. An additional evaluation of the clinical specimens obtained from different anatomical areas can affect the optimal strategy of treatment prescription [40,41].

The high prevalence of *M. genitalium* mutations associated with the resistance to macrolide, fluoroquinolone, or both agents in the examined population of Moscow City patients allows us to conclude that the detection of biomarkers indicating *M. genitalium* resistance to these agents should be included in the routine procedure of STD screening in conjunction with the detection of the pathogen in general. Besides the optimization of patient diagnosis algorithms, it is very important to improve pathogen eradication therapy regimens to restrain the development and transmission of resistance.

## 5. Limitations of the Investigation

The lack of data on the clinical outcomes or therapy adjustment for the examined patients does not allow us to make a convincing conclusion about the correlations between the reported mutations and treatment efficacy. It is also impossible to determine the exact paths of infection transmission among the patients because there is no information on re-infection, pathogen persistence, or the duration of infection. Without thorough genotyping of the isolates, it is impossible to understand whether *M. genitalium* infection could be persistent or recurrent due to treatment failure or re-infection. The advanced aspects of the connection between in vitro resistance and clinical outcomes may be determined in large-scale clinical studies. Taking the increase in treatment resistance into account, the international guidelines propose to perform routine testing of all *M. genitalium*-positive samples for the determinants of therapy resistance.

## 6. Conclusions

The inefficiency of the recommended therapeutic strategies observed by clinicians over the last few years have resulted from the increasing resistance of *M. genitalium* to macrolide and fluoroquinolone antibiotics. The high prevalence of isolates with markers of resistance to these two groups of therapeutic agents suggests that the existing treatment strategies should be adjusted to improve the response to therapy and complete pathogen eradication. The implementation of combined diagnostic tests capable of simultaneous pathogen detection and resistance biomarker identification is extremely important for the improvement of first-line treatment efficacy and the reduction of the burden placed on the healthcare system. The latter goal would be achieved through the dismissal of recurrent patient examinations and inefficient administration of antimicrobial agents for the treatment of *M. genitalium*-positive patients. Our results also support the idea of focused screening of patients with chlamydial infections for *M. genitalium* presence as an economically advantageous strategy. The data generated in this study emphasize the importance of constant epidemiologic surveillance of the resistance dynamics in *M. genitalium* aimed at restraining its development and spread.

## Figures and Tables

**Figure 1 pathogens-12-00496-f001:**
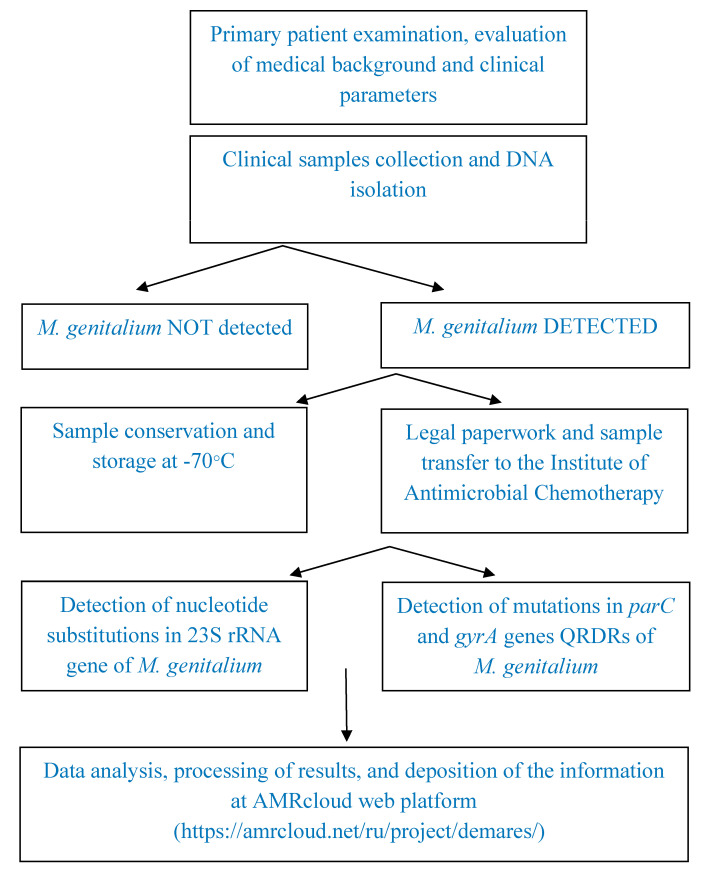
The steps for the evaluation of macrolide and fluoroquinolone resistance markers’ prevalence in *M. genitalium*.

**Table 1 pathogens-12-00496-t001:** Oligonucleotide primers used for the detection and characterization of mutations in the 23S rRNA and *parC* and *gyrA* genes of *M. genitalium*.

Primer Name	Primer Sequence, 5′-3′
Mpg gyrA SeqF	CCTGATGCTAGAGATGGACTTAAACC
Mpg gyrA SeqR	TAATCTTGCTTCTGTATAACGTTGTGC
Mpg parC SeqF Mpg parC SeqR	GTCTTTGCAGTTAGCTTTAGTAAGTATGC CTCGCACCATTGATAAAGAGGTTAGG
Mpg 23s SeqF Mpg 23s SeqR	CGTCCCGCTTGAATGGTGTAAC GCGCTACAACTGGAGCATAAG

**Table 2 pathogens-12-00496-t002:** Frequency of mutations in the *M. genitalium* 23S rRNA gene and QRDRs or *parC* and *gyrA* genes in the samples from the examined patient population.

23S rRNA Gene of *M. genitalium*	N = 213	%	95% CI
Wild-type (S)	158	74.2	67.9–79.6%
Mutations (R)	55	25.8	20.4–32.1%
A2059G	36	65.5	52.3–77%
A2058G	19	34,6	23.4–47.8%
***par*C gene**	**N = 213**	%	95% CI
Wild-type (S)	176	82.6	77–87.1%
Mutations (R)	37	17.4	12.9–23.1%
D84N (Asp87Asn) ^a^	20	54.1	38.4–69%
S80I (Ser83Ile)	12	32.4	19.6–49%
S80N (Ser83Asn)	3	8.1	2.8–21.3%
D84G (Asp87Gly)	1	2.7	0.5–14%
D84Y (Asp87Tyr)	1	2.7	0.5–14%
***par*C+23S rRNA gene**	**15**	7.1	4.3–11.3%

^a^*M. genitalium* numbering.

## Data Availability

Not applicable.

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
