# Peer review of "Genetic Determinants of Macrolide and Fluoroquinolone Resistance in Mycoplasma genitalium and Their Prevalence in Moscow, Russia"

_pathogens, 2023, doi:10.3390/pathogens12030496_

Round 1

Reviewer 1 Report

The authors present an interesting study about Mycoplasma genitalium and presence of the drug resistance genes in an institutional study in Mosow. I have following comments upon the study.

(A) Details of institutional ethical clearances are not included, therefore, must be mentioned in methods.

(B) Please mention as to why a combination of 16s along with 23s is not used in the discussion section.

(B) 

Author Response

Dear reviewer, thank you very much for your careful evaluation of our article and your valuable comments. We have written our responses in the file.
Regards,

Inna Edelstein

Reviewer 2 Report

The manuscript submitted by Edelstein et al. summarizes results of a study investigated resistance-associated mutations in Mycoplasma genitalium strains among patients of a specialized center in Moscow, Russia. Against the background of limited therapeutic options and increasing rates of acquired resistance, further data regarding the prevalence of resistant strains in different populations are of importance. Overall, the paper is informative and well-written.

Comments:

1.     Use of E. coli numbering to describe mutations in parC gene and ParC protein is correct but unusual internationally. For better understanding, please include also M. genitalium numbering if mutations are characterized for the first time.

2.     It remains unclear for me why “… the absence of a cell wall dictates the usage of molecular biology techniques…”. Explain or re-write.

3.     It was stated that resistance in M. genitalium “usually” originates from mutations of target genes. This suggests that further mechanisms are known which is not the case.

4.     Experimental design: Selection of patients is unclear. It was mentioned that screening of asymptomatic patients for M. genitalium is not meaningful. What are the criteria for including patients in the study? Do you have any information about symptoms of patients? Prevalence of M. genitalium as well as of resistance-associated mutations are higher in risk populations. Are there any information about MSM among the persons included in the study? Please add a short description of the method for detection of Chlamydia trachomatis.

5.     Results (3.2.): You don’t have information that strains with wild-type 23S rRNA “…display normal sensitivity to macrolides…”. Please delete or re-phrase. What is meant with “…impact…”? There are discrepancies between text and table 2 (36 vs. 37 strains with mutations, 19 vs. 20 strains with D84N substitution). Please correct.

6.     Discussion: Is the mentioned guideline recommending the josamycin treatment a national directive? In international guidelines (like the European) a strict regime of first (azithromycin), second (moxifloxacin) and third antibiotics (pristinamycin, minocycline) for treatment of Mg infections is defined. Please discuss in more detail.

I miss a placement of your data into the results of international studies. As relatively actual reviews and meta-analyses exist (e.g., Fernandez-Huerta et al., 2020, Machalek et al., 2020), add some sentences.

To my understanding, statement in sentence “In comparison to data obtained in 2019…” is in relation to a previous study from Russia (Moscow?). This should be mentioned. Is reference [29] really the right one?

The intensive discussion of ofloxacin treatment in the section starting with “Poorly controlled…” is unclear as this antibiotic is not recommended for treatment of Mg infections. Is ofloxacin still intensively in use for these infections in Russia or not? Please discuss and consider shortening of section.

Without typing results, a “…clonal expansion…” of resistant strains cannot be substantiated and spread might also be caused by different clones. Please re-write.

I miss your recommendation for treatment of patients carrying a strain showing mutations associated with macrolide and quinolone resistance under the conditions in Russia. Please add.

Why should the detection of Mg from “…different anatomical areas…” affect the treatment strategy? Please explain.

7.    Conclusions: The conclusion to screen Chlamydia-positive patients for M. genitalium remains unclear. According to your data, 38% of positive persons had a co-infection with Mg and the remaining not.

Author Response

Dear reviewer, thank you very much for your thorough analysis of our work and for your important comments and suggestions on how to improve the article. We have made all corrections to the text and provided our responses in the attachment.
Regards,
Inna Edelstein

Reviewer 3 Report

Review Report Manuscript ID pathogens-2229884

Congratulations on the present study! The original manuscript needs to be improved for publication acceptance. Please see my comments:

General/Major comments

The original manuscript does not possess uniformity in the text, such as observed in the abstract section, and neither the document shows continuous lines to facilitate the revision. Please rectify it in the revised version of the manuscript.

The abstract seems to be long for the maximum number of words allowed by the Pathogens journal, please check it in the revision.

Although the manuscript is well-written and no major comments are needed, the text needs some English Editing, for example, the authors have INTRODUCTION, Materials and Methods, RESULTS, DISCUSSION, and Conclusion as section titles. Please maintain the same formation.

Minor comments

Introduction

Page 2- Please state the full names before the abbreviation “WHO”.

Page 2- Please replace “The causal treatment of mycoplasma infection …” with “The causal treatment of mycoplasmal infection …”.

Page 2- Please rectify the references “[16,17.”.

Page- Please add the proper references to the following entire paragraph without citations: “Therapy failure is usually linked to the primary infection with strains that have al-ready developed a resistance-associated mutation profile due to previous antibiotic expo-sure (poor personal compliance of the patient, lack of full recovery confirmation) or to the treatment-induced development of resistance, that is further transmitted through sexual contacts. The screening for M. genitalium infection is not performed in a routinely among asymptomatic patients, and the undefined microbial status may contribute to the spread-ing of mutant strains. In the case of co-infection with Chlamydia trachomatis, the empirical therapy with antibiotics can further boost the resistance development and spreading, be-cause the antibiotic agents are prescribed with the prospect of concurrent elimination of two pathogens at once.”.

Page 2-3- The aim goal should be put at the end of the previous paragraph and it should not be as a paragraph by itself: “AIM: To evaluate the …”. Please change it.

Materials and Methods

Page 3- Please state the full names before the abbreviation “MSPCDC”.

Page 3- Please briefly describe the main procedure of “M. genitalium DNA presence was detected using “AmpliPrime® NCMT” PCR reagent kit (NextBio LTD, Moscow, Russia)”.

Page 4- Please add the Human Committee Approval Number at subsection 2.2.

Page 4- Please add the source or previous publication of the applied primers in Table 1.

Results

Page 4-5- The values with decimals should be written with dots and not a comma, some examples “Median age was 32,3 years …”, “… M. genitalium samples (74%; CI: 67,9-79,6%) …”, and values in Table 2.

Table 2- Please put “M. genitalium” in italics.

Discussion

Page 7- Please add references: “One of the plausible causes for the increase in the number of samples with such a genotype is a selective pressure from macrolide and fluoroquinolone therapy due to the extensive use of these agents in both in-patient facilities and ambulatory treatment in recent years, especially in conjunc-tion with anti-COVID-19 therapy. The empirical prescription of these agents without an opportunity for timely STD diagnosis and the lack of therapy efficacy checks via confirm-atory tests at the same time could promote the clonal expansion of such strains. A reason-able switch to fluoroquinolone therapy in the case of poor efficacy of macrolides provided ground for the development of the secondary resistance to the new family of antibiotics and contributed to stabilization and circulation of such isolates in the population. Consid-ering the statistically significant increase in the fraction of strains with such genotypes, the use of alternative therapeutic regimens or the introduction of new antimicrobial agents is necessary to overcome this spreading resistance. The definitive proof of a causal bond between different types of mutations identified in clinical M. genitalium samples and ther-apy failure may be obtained through the development of in vitro approaches to the evalu-ation of clinical isolates’ sensitivity to antimicrobial agents using classic culture tech-niques.

All proposed therapy variants should be based on efficient synchronous C. tracho-matis and M. genitalium eradication and exclude monotherapy with doxycycline or one-time administration of azithromycin. Unlike C. trachomatis, M. genitalium quickly develops resistance to the antibiotics of both first- and second-line treatment due to only one ribo-somal operon being present in the genome. Our data indicate that M. genitalium can al-ready manifest fluoroquinolone resistance prior to initial macrolide treatment and vice versa, therefore the ideal diagnostic approach should include testing for biomarkers asso-ciated with resistance to both agent groups.”

Congratulations again on the present study! However, to be honest, the present study is more suitable as a brief report or short communication than as a full-length article. Nonetheless, I consider the study relevant to the scientific community and I recommend minor revisions for publication acceptance.

Author Response

Dear reviewer, we thank you for your careful attention to our work, your careful data checking and your pleasant review of our research. We have made all corrections and attached a full description of all additions in the attachment.
Regards,
Inna Edelstein
